# Morphometric Assessment of the Hip Joint in a Functional Dysplastic Rabbit Model

**DOI:** 10.3390/vetsci11080387

**Published:** 2024-08-22

**Authors:** Inês Tomé, Luís Costa, Sofia Alves-Pimenta, Roberto Sargo, José Pereira, Bruno Colaço, Mário Ginja

**Affiliations:** 1Veterinary Teaching Hospital, University of Trás-os-Montes e Alto Douro, 5000-801 Vila Real, Portugal; inestome@utad.pt (I.T.);; 2Associate Laboratory for Animal and Veterinary Science (AL4AnimalS), CECAV-Veterinary and Animal Science Research Centre, University of Trás-os-Montes e Alto Douro, Quinta de Prados, 5000-801 Vila Real, Portugal; 3Department of Veterinary Sciences, University of Trás-os-Montes e Alto Douro, 5000-801 Vila Real, Portugal; 4Department of Animal Science, University of Trás-os-Montes e Alto Douro, 5000-801 Vila Real, Portugal

**Keywords:** hip dysplasia, hip joint, imaging morphometry, surgically induced model, osteoarthritis, rabbit

## Abstract

**Simple Summary:**

Hip dysplasia is an orthoedic condition where the hip joint does not develop correctly which leads to pain and discomfort in the human and canine species. To better acknowledge this condition and find new treatment modalities, a model of hip dysplasia performed in rabbits by surgically modifying the hip joint was presented. A total of seventeen rabbits were used to assess morphological changes in the hip joint using radiography and computed tomography. Our research revealed that the surgery caused significant changes in the hip joint, making it similar to the hip dysplasia seen in dogs and humans. These changes included the increased rotation of the femur and a shallower acetabulum, which are key features of hip dysplasia. This rabbit model can be used to evaluate the progress of hip dysplasia and test new treatments, which may lead to better prevention and treatment options for hip dysplasia and the improvement of the quality of life for both animals and humans affected by this condition.

**Abstract:**

The present study investigates the morphometric changes in the hip joint in a surgically induced rabbit model of hip dysplasia through the sectioning of the ligamentum capitis femoris and pelvic limb immobilization. A total of seventeen rabbits were evaluated using radiographic and computed tomographic imaging to measure the following parameters: the femoral angles of anteversion and inclination, length and width indexes of the neck of the femur, and acetabular depth and ventroversion. Significant differences in femoral anteversion angle and acetabular depth were observed, particularly in the group of hip instability surgery with pelvic limb immobilization. The results have shown the influence of hip joint instability in the promotion of femoral anteversion and acetabular shallowing. These findings provide a foundation for future research on naturally occurring or experimentally induced hip dysplasia in rabbits and underscore the model’s potential for studying the biomechanical and developmental aspects of hip joint disorders.

## 1. Introduction

Rabbits, since the 1960s, have been extensively used as animal models to study hip dysplasia (HD) and are also commonly employed in additional research [1,2]. Initially, these models served solely to assess the disease’s pathophysiology due to the lack of the functionality of the intervened limbs [1,3,4]; however, a recent study successfully induced HD, maintaining limb functionality while providing valuable medical inputs in the research therapeutics of HD. This model involved the surgical sectioning of the ligamentum capitis femoris [5].

The effect of such models could be determined by comparing the morphometric data of the pelvic limb. Unfortunately, the comprehensive imaging morphometry of the pelvic limbs in both healthy and hip dysplastic rabbits remains poorly documented [4,6]. In young rabbits, the head of the femur and acetabulum form a three-dimensional anatomical structure whose development is directly interrelated with the onset of HD [7]. It is widely acknowledged that the acetabulum’s concave shape and depth depend greatly on the presence and pressure of the head of the femur [8]. Moreover, the angles of the inclination and anteversion of the femoral neck are closely related to the magnitude and location of pressure between these anatomical structures [9,10]. Also, the epiphyseal growth plate of the proximal femoral neck is a dynamic entity susceptible to the mechanical influence of compressive loads [11].

Our hypothesis is that the surgical sectioning of the ligamentum capitis femoris results in hip instability and changes the anatomical pressure between the head of the femur and the acetabulum, leading to developmental abnormalities in both structures. To the authors’ knowledge, no published studies have provided detailed radiographic and computed tomographic morphometric measurements of biomechanical interest in rabbit hip joints, such as the femoral angles of anteversion (FAA) and inclination (FAI), length and width indexes of the neck of the femur (FNLi and FNWi), acetabular depth index in X-ray (ADi_xR) and in the computed tomography (ADi_CT), and acetabular ventroversion (AV). The main aim of this study was to quantify these radiographic and computed tomographic parameters in healthy and surgically induced rabbit hip joints. These data may allow comparisons between the morphometric changes observed in rabbit hip joints and those seen in the human and canine species, which are also prone to HD development.

## 2. Materials and Methods

### 2.1. Animal Model

This study utilized seventeen healthy male New Zealand white rabbits, aged 6 weeks old (*n* = 34 hips). The animals were sourced from La Granja San Bernardo and were housed in cages sized 55 cm × 80 cm × 55 cm with an internal platform, at a relative humidity of 50–60%, temperature of 19–20 °C, and exposed to a 12-h light–12-h dark cycle. The rabbits were subject to a quarantine of 10 days before the beginning of this study and were provided with a standard pellet diet with water provided ad libitum. All the procedures complied with the European and National legislation on the protection of animals used in scientific research (European Directive 2010/63/EU and National Decree-Law 113/2013) and received approval from the competent Portuguese authority, the Directorate-General for Food and Veterinary (DGAV_0421/000/000/2022).

### 2.2. Study Design

The rabbits were randomly assigned to three groups: Group I (GI), the control group with 3 rabbits; Group II (GII), the group subject to the instability surgery on the left hip joint; and Group III, the group in which an instability surgery was performed in the left hip joint, followed by left stifle fixation in extension with appropriate medical dressings for 3 days to promote the hip dysplasia development in 7 rabbits.

For research purposes, the hip joints were classified into normal hip joints, NH (*n* = 6 hips), the left and right hip joints of GI; instability surgery hip joints, ISH (*n* = 7 hips), the left hips of GII; sham surgery hip joints, SSH (*n* = 7 hips), the right hip joints of GII; instability surgery hip joints with the pelvic limb immobilization, ISIH (*n* = 7 hips), the left hip joints of GIII; and hip joints without surgery, HWS (*n* = 7 hips), the right hip joints of GIII.

### 2.3. Hip Joint Surgery and Bandage Immobilization of the Pelvic Limb

The hip joint surgery was performed using a ventral inguinal access, by exposing and sectioning the joint capsule. Later, the ligamentum capitis femoris was sectioned, and ventral subluxation was promoted. The pelvic limb immobilization in GIII was accomplished by a soft padded bandage applied over the pelvic limb from toes to mid-femur and was maintained for 3 days. Costa L. et al. 2024 described the surgery procedure in detail [5].

### 2.4. Imaging Morphometry

Fourteen weeks post-surgery, the rabbits underwent radiographic and computed tomographic studies under general anesthesia. The pre-anesthetic protocol included butorphanol (Butomidor^®^, Richter Pharma AG, Wels, Austria, at 0.4 mg/kg, IM) and dexmedetomidine (Sedadex^®^, Le Vet Beheer B.V., Oudewater, The Netherlands, at 50 μg/kg, IM). Aesthesia induction was achieved using ketamine (Ketamidor^®^, Richter Pharma AG, Wels, Austria, at 20 mg/kg, IM) combined with midazolam (Dormazolan^®^, Le Vet Beheer B.V., Oudewater, The Netherlands, at 0.5 mg/kg, IM). The maintenance of volatile anesthesia was performed with 1.5% isoflurane (IsoFlo100%^®^, Zoetis, Lisbon, Portugal) in oxygen using a facial mask. All the measurements were conducted using the free DICOM Medical Image Viewer Software (Horos, version 4.0.0 RC5).

#### 2.4.1. Radiographic Assessment

The animals were radiographed using an X-ray machine (Optimus 80, Philips, Best, The Netherlands) and a digital radiography Fujifilm detector (Fujifilm, New York, NY, USA). A ventrodorsal hip joint distraction and extended view, and mediolateral femoral view were performed to evaluate the FAA, FAI, FNLi, FNWi, and ADi_xR.

FAA was measured using the biplanar method (Reynold’s technique) and determined by the tangent of the distances ‘a’ and ‘b’ (tan (a/b), in mm) (Figure 1). Distance ‘a’ was assessed in the mediolateral radiograph and consists of the perpendicular distance between the center of the best-fit circumference drawn in the outer limit of the head of the femur and the axis of the proximal diaphysis of the femur defined by the center of the two best-fit circumferences at the most distal diaphysis between the greater and third trochanters and the most distal diaphysis of the proximal femur. Distance ‘b’ was measured in the ventrodorsal radiograph and entails the perpendicular distance between the center of the best-fit circumferences drawn in the outer limit of the head of the femur and between the greater and third trochanters and the femoral axis defined by the center of the two best-fit circumferences drawn between the most proximal and the most distal diaphysis of the femur [11,12]. FAI is the angle formed by the line connecting the center of the best-fit circumferences drawn at the head of the femur and between the greater and third trochanters and the line connecting the center of the best-fit circumferences at the most proximal and distal diaphysis border of the femur [11].

FNLi is defined as the ratio of the length of the neck of the femur (distance ‘l’) and the diameter of the head of the femur (l/Ø, in mm) (Figure 2) [12]. The length of the neck of the femur was defined as the distance between the outer limit of the best-fit circumference drawn in the head of the femur and the center of the best-fit circumference drawn between the greater and third trochanters. The FNWi was measured as the ratio between ‘w’, the width of the neck of the femur, and the distance ‘l’ (w/l, in mm). Distance ‘w’ was established as a line perpendicular to ‘l’ and at the narrowest distance of the neck of the femur tangent to the outer limit of the best-fit the circumference between the greater and third trochanters [13].

ADi_xR was evaluated as the ratio between ‘d’, the acetabular depth, and the acetabular diameter (d/Ø, in mm) (Figure 3). Distance ‘d’ was defined as the biggest perpendicular distance concerning the line tangent to the craniolateral and caudal acetabular edges [12].

#### 2.4.2. Computed Tomographic Assessment

The rabbits were submitted to a computed tomography (CT) assessment (Revolution™ ACT-16 slices, General Electric Medical Systems, Buckinghamshire, UK) and were positioned in sternal recumbency in a knee weight-bearing position in a hole of foam sponge. It was defined as a field-of-view that covered the entire hip joint and femur, and the images were acquired in the helical mode using a thickness of 1.25 mm and bone algorithm reconstruction (WL 400, WW 2000). The measurements performed were the ADi_TC and AV (Figure 4).

ADi_CT was measured as the ratio between ‘d’, the acetabular depth, and the acetabular diameter (d/Ø, in mm). Distance ‘d’ was defined as the biggest perpendicular distance to the line tangent to the dorsal and ventral acetabular edges [12]. The AV was measured as the angle between the line that intersected the center of the coccygeal vertebrae body and the middle of the pubic symphysis and another line tangent to the most lateral edges of the dorsal and ventral acetabular rim [13].

### 2.5. Statistical Analysis

The data were analyzed using the SPSS Statistics software for Windows (Version 27.0, IBM, Armonk, NY, USA). The Shapiro–Wilk test was used to evaluate the normality of the values measured. Non-normally distributed variables were assessed by the Kruskal–Wallis test and Dunn’s multiple comparisons test, and were summarized using the median and interquartile range. The statistical significance was set at *p* < 0.05.

## 3. Results

A total of seventeen rabbits, fourteen weeks post-surgery, were included in the imaging assessment (*n* = 34 hip joints), presenting a mean body weight of 4.34 ± 0.38 kg (mean ± standard deviation). No animal revealed locomotion comorbidities after the removal of the soft padded bandage.

In the FAA assessment, NH showed the retroversion of the neck of the femur with a median of 10.6°. The HWS revealed the lowest negative FAA median (−15.48°), followed by NH, HWS, and ISH. The ISHI was the only group that showed a positive median value (+3.24°) (Table 1). In the Kruskal–Wallis test, significant differences were found between the groups.

The FAI assessment showed very similar angles in all the hip joint groups. The HWS had the highest median angle (113.37°), followed by ISHI, SSH, NH, and ISH, with medians only showing slight variations among the groups, which is the reason why FAI did not evidence significant differences between the research hip joint groups in the Kruskal–Wallis test.

NH group had the highest median value of 1.14 in the FNLi measurements, followed by ISHI, ISH, SSH, and HWS, and the groups revealed a quite wide interquartile range, without statistically significant differences in the Kruskal–Wallis test.

ISHI exhibited the highest median of 0.56 in the FNWi, followed by ISH, HWS, SSH, and NH. The width values of the neck of the femur reflected a consistent tendency across the groups that despite the absence of statistical significance indicated a higher median value of FNWi for the surgically induced groups and a lower median value for the other groups.

In both ADI_xR and ADi_CT, NH revealed the highest median values of 0.59 and 0.27, respectively, and the ISH and ISHI groups showed the lowest median values. Only in the ADI_xR assessment, statistically significant differences were observed between the groups evaluated in the Kruskal–Wallis test.

In the AV evaluation, the group NH demonstrated the highest median of 23.29° in the AV assessment, followed by ISH, ISHI, SSH, and HWS. No statistical differences were observed between the groups evaluated in the Kruskal–Wallis test.

The Kruskal–Wallis test and Dunn’s multiple comparison test were used to evaluate which groups showed statistically significant differences in the FAA and Adi_xR assessments. The FAA analysis revealed statistically significant differences when comparing ISH with HWS and when comparing ISHI with NH, SSH, and HWS. In the ADi_xR evaluation, statistical significance was found when comparing ISHI with NH, SSH, and ISH (Table 2).

## 4. Discussion

Rabbits have been extensively employed in biomedical research to study various aspects of HD, namely femur and acetabular morphology, and physiopatogeny [1]. Some studies have shown that the sensory innervation of the hip joint seems to play a relevant role in assessing clinical signs, which may be beneficial in therapeutic approaches [2,14].

The morphometric changes in the hip joint associated with HD have been studied in both human and veterinary medicine, and different types of acetabular and femoral HD have been described [12,14,15]. Our results agree with previous studies that consider the hip joint as a dynamic entity, whose morphology is closely dependent on the mechanical influence in the course of its development [9,10,16]. The animals used in experimentally induced models must be young enough so that their growth plates in the proximal femur and acetabulum are still open and able to respond to new existing loads [11] as it occurs in the development of naturally occurring HD. It should be added that relatively little attention has been given to the morphometric changes in the hip joints of rabbits when applied in HD experimental research [17]. This species is commonly used as an HD model, allowing a foreseeable translation of findings in bioengineering studies concerning the disease in humans and dogs [4,18]. In rabbits, the radiographic signs of spontaneous hip joint osteoarthritis have been documented in adult animals over one year of age [18]. Given the young age of the animals in our sample, around 20 weeks old, it is very unlikely that the imaging signs of hip joint osteoarthritis recorded are due to natural occurrence.

The FAA measured in this study in the normal hip joints (NH group) has a very different range in comparison with the FAA observed in humans and dogs, in which it exhibited anteversion [12,19]. In rabbits, the head of the femur showed retroversion in all the hip joints of the NH group, hip joints without signs of HD. This conformation is directly related to the evolutionary optimization of functionality and the particularities that this species has regarding pelvic limb biomechanics and gait patterns [20]. The rabbit rests in sternal decubitus with the pelvic limbs flexed, slightly abducted, and supinated, and adopts a bounding or half-bounding gait in which the pelvic limbs push off and move simultaneously [20,21], with the greatest forces occurring in the initial stance phase [20]. The medical bandage applied in the ISHI group promoted the knee joint in an extended position and tightened the hamstring, resulting in the hip capsule being stretched, which was amplified by the absence of the ligamentum capitis femoris and led to subluxation or dislocation [22]. As a result, the ligamentum capitis femoris appears to be crucial as a head of the femur–acetabulum stabilizer, especially when the limb has been immobilized in extension [22,23]. The head of the femur dislocation contributes to the drift in load-bearing forces acting on the growth plate of the neck of the femur, possibly compressing its most cranial region too heavily, inhibiting its development, and leading to the growth of its caudal region and directing the neck of the femur cranially. Previous works also confirmed changes in the degree of the anteversion of the neck of the femur in the cases of the subluxation and inhibition of plate growth chondrocyte proliferation due to the excess load [11,12].

The absence of significant differences between the groups in the FAI, FNLi, FNWi, and AV assessments may show that the surgical model presented did not favor or neglect different joint pressures in some locations. In the case of the FAI assessment, the absence of statistical differences may be due to the fact that the model used did not have a decisive influence on the different pressures applied at the dorsal and ventral aspects of the growth plate of the neck of the femur. Given the expected dynamics, greater ventral or dorsal pressure could result in a reduction or increase in the FAI values, respectively. Furthermore, when determining FAI, the methodology previously described for the dog was not able to be applied entirely [12] due to anatomical conformation differences in the rabbit femur, namely the position of the femur that is horizontal in the rabbit, oblique cranioventral in the dog [24] and vertical in humans [25] and the presence of a third trochanter absence in both humans and dogs [26]. As a result, the definition of the axis of the neck of the femur based solely on the center of the proximal circle of the femoral diaphysis resulted in a line that did not match with the center of the axis of the neck of the femur in the rabbit. The AV angle is directly subject to the dorsal and ventral growth of the acetabular bone structure [13]. Since any pressure variation in this location can only be exerted by the head of the femur, the absence of developmental changes follows the results described in the FNLi assessment. The absence of changes in the FNLi may be associated with the overload pressure at the growth plate area that did not have a significant effect and the normal growth in the other areas was sufficient to counterbalance it.

The depth and shape of the acetabulum develop in response to the presence of the sphericity of the head of the femur and due to the growth of the internal acetabular cartilage and chondrocyte proliferation at the acetabular margin [8,10]. The ADi_xR is measured as a line tangent to the cranial and caudal acetabular borders [12] and, as already discussed previously in the FAA assessment, the results were compatible with an overload of the cranial aspect of the head of the femur, whose contact forces were transmitted to the cranial acetabular edge. Therefore, we hypothesized that this increase in pressure also interferes with the development of the most cranial portion of the acetabulum, leading to a shallower acetabular depth. These results are in line with previous research carried out in dogs [12] and humans [27]. Regarding the ADi_CT, this parameter depends on dorsal and ventral acetabular development [13], as observed with the following variables, FAI and AV. In the ADi_CT assessment, group HWS showed greater heterogeneity between the hips evaluated.

Rabbits, dogs, and humans share similarities in the hip joint anatomy and development, making the rabbit a useful model for translational research. The differences in FAA between rabbits, dogs, and humans highlight the unique evolutionary adaptation in each species, while the acetabular development underscores common biomechanical influences across species. Using this rabbit model, researchers can study the progression of HD, test potential surgical interventions, and evaluate new therapeutic approaches that may apply to other species. Thus, this study not only contributes to our understanding of HD but also provides a foundation for a broader comparative analysis and translational applications to both veterinary and human medicine.

The present work has several limitations. Rabbits, despite being frequently used as a hip joint osteoarthritic model, require a careful extrapolation of data to dogs or humans due to the differences in the pelvic limb biomechanics and gait. Moreover, some results of the study may have been compromised due to our small sample size. Additionally, imaging morphometry is associated with inaccuracies arising from two main sources, the different positioning of patients and the inaccurate locations of the landmarks in images, which may be responsible for the wide interquartile range between the groups.

## 5. Conclusions

This study demonstrated the critical role of the ligamentum capitis femoris sectioning in joint development, where its absence led to the subluxation or dislocation of the head of the femur and the consequent joint overload. The resulting contact between the cranial portion of the head of the femur and the acetabulum promotes femoral anteversion and the shortening of the cranial edge of the acetabulum. These findings highlight the value of this rabbit model for studying HD and its potential translational relevance to other species, including humans and dogs.

## Figures and Tables

**Figure 1 vetsci-11-00387-f001:**
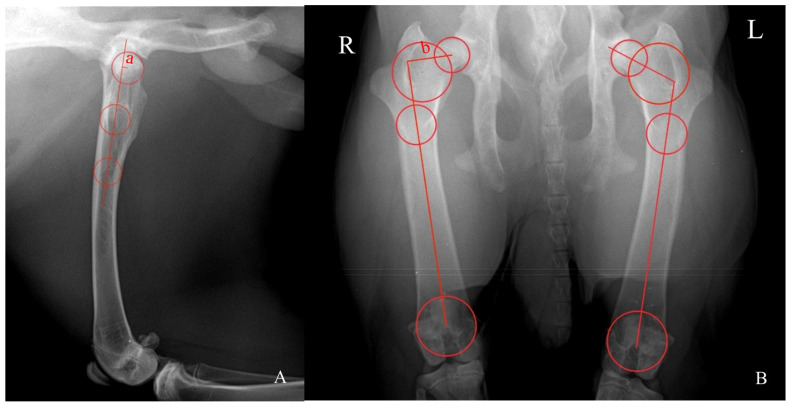
Illustration of the methodology used for assessing the anteversion (FAA) and inclination (FAI) angles of the head of the femur. (**A**) The mediolateral view of the hip joint. (**B**) The ventrodorsal view of the hip joint. R: right side. L: left side. FAA was determined by the tangent of the distances ‘a’ divided by ‘b’. Distance ‘a’ (**A**) consists of the perpendicular distance between the circumference’s center drawn at the head of the femur and the axis of the proximal femoral diaphysis defined by the circumferences’ center at the most distal diaphysis between the greater and third trochanters and the most distal diaphysis of the proximal femur. Distance ‘b’ (**right side** of (**B**)) entails the perpendicular distance between the circumferences’ center drawn in the head of the femur and between the greater and third trochanters and the femoral axis defined by the circumferences’ center drawn between the most proximal and the most distal diaphysis of the femur. FAI (**left side** of (**B**)) is the angle formed by the line connecting the circumferences’ center drawn at the head of the femur and between the greater and third trochanters and the line connecting the circumferences’ center at the most proximal and distal diaphysis border of the femur.

**Figure 2 vetsci-11-00387-f002:**
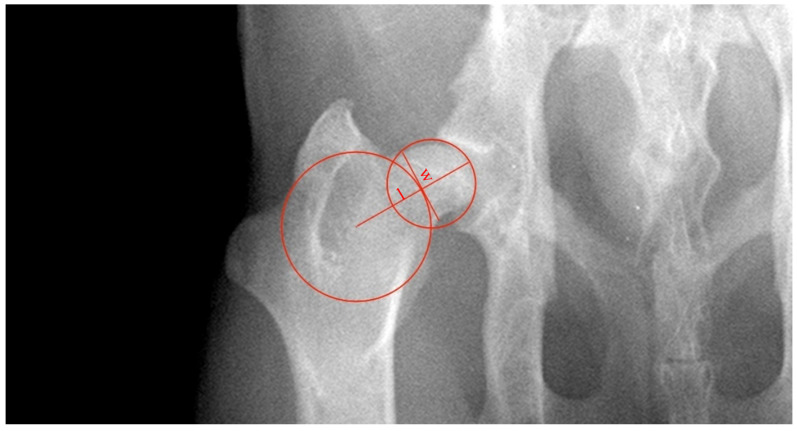
Illustration of the methodology used to assess the length index of the neck of the femur (FNLi) and femoral neck width index (FNWi) in a ventrodorsal radiograph of the hip joint. FNLi is calculated as the ratio of ‘l’, the length of the neck of the femur, and the diameter of the head of the femur. The distance ‘l’ was defined as the length between the outer limit of the circumference drawn in the head of the femur and the circumference’s center drawn between the greater and third trochanters. The FNWi was measured as the ratio between ‘w’, the width of the neck of the femur, and the distance ‘l’. Distance ‘w’ was established as a line perpendicular to ‘l’ and at the narrowest distance of the neck of the femur tangent to the outer limit of the circumference between the greater and third trochanters.

**Figure 3 vetsci-11-00387-f003:**
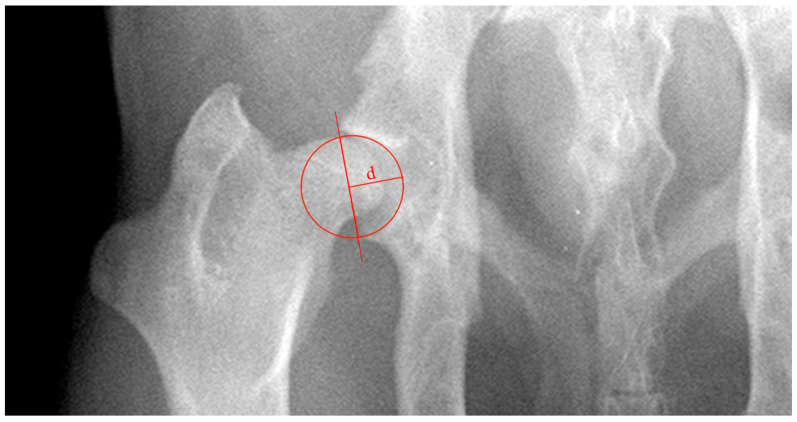
Ventrodorsal radiograph of the hip joint illustrating the methodology used to evaluate the acetabular depth index (ADi_xR). ADi_xR was evaluated as the ratio between ‘d’, the acetabular depth, and the acetabular diameter. Distance ‘d’ was defined as the biggest perpendicular distance concerning the line tangent to the craniolateral and caudal acetabular edges.

**Figure 4 vetsci-11-00387-f004:**
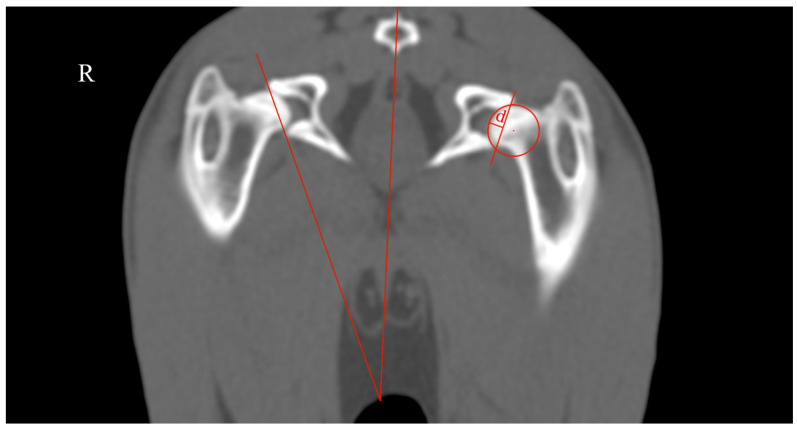
Transverse CT scan of the hip joint illustrating the methodology used to evaluate the acetabular depth index (ADi_CT) and the acetabular ventroversion (AV). R: right side. ADi_CT (**left side**) was measured as the ratio between ‘d’, the acetabular depth, and the acetabular diameter. Distance ‘d’ was defined as the biggest perpendicular distance to the line tangent to the dorsal and ventral acetabular edges. The AV (**right side**) was measured as the angle between the line that intersected the center of the coccygeal vertebrae body and the middle of the pubic symphysis and another line tangent to the most lateral edges of the dorsal and ventral acetabular rim.

**Table 1 vetsci-11-00387-t001:** Descriptive statistics of the data presented as median (quartile 25–75%) of the radiographic and computed tomographic assessment in 17 rabbits (*n* = 34 hips).

	NH	ISH	SSH	ISHI	HWS
*n* (number of hips)	6	7	7	7	7
Anteversion Angle Median (Quartiles 25–75%) (in degrees)	−10.60 [−17.35–(−6.91)]	−7.43 (−12.63–0.73)	−9.07 [−15.53–(−6.59)]	3.24 (−9.93–17.56)	−15.48 [−17.29–(−10.52)]
Inclination Angle Median (Quartiles 25–75%) (in degrees)	110.51 (109.39–115.56)	109.82 (103.50–116.21)	111.20 (109.6–113.32)	112.30 (107.42–116.34)	113.37 (110.49–118.91)
Femoral Neck Length Index Median (Quartiles 25–75%)	1.14 (1.10–1.18)	1.10 (1.08–1.14)	1.07 (1.04–1.10)	1.13 (1.08–1.16)	1.03 (1.03–1.16)
Femoral Neck Width Index Median (Quartiles 25–75%)	0.45 (0.44–0.47)	0.52 (0.48–0.57)	0.49 (0.44–0.51)	0.56 (0.46–0.62)	0.50 (0.44–0.51)
Acetabular Depth Index—X-ray Median (Quartiles 25–75%)	0.59 (0.53–0.63)	0.53 (0.49–0.57)	0.56 (0.54–0.60)	0.38 (0.37–0.47)	0.53 (0.52–0.55)
Acetabular Depth Index—CT Median (Quartiles 25–75%)	0.27 (0.24–0.29)	0.20 (0.18–0.27)	0.22 (0.17–0.26)	0.21 (0.16–0.28)	0.26 (0.21–0.27)
Acetabular Ventroversion Angle Median (Quartiles 25–75%) (in degrees)	23.29 (19.79–25.36)	21.02 (16.06–23.99)	21.03 (17.00–27.18)	21.81 (14.41–23.11)	17.59 (16.79–24.53)

NH: normal hip joints; ISH: instability surgery hip joints; SSH: sham surgery hip joints; ISIH: instability surgery hip joints with pelvic limb immobilization; HWS: hip joints without surgery.

**Table 2 vetsci-11-00387-t002:** Summary of Kruskal–Wallis test followed by Dunn’s multiple comparisons test of the femoral anteversion angle (in degrees) and acetabular depth index—X-ray in 17 rabbits (*n* = 34 hips).

	Femoral Anteversion Angle (in Degrees)				Acetabular Depth Index—X-ray			
Group a-Group b	Test Statistic	Standard Error	Standard Test Statistic	Significance *	Test Statistic	Standard Error	Standard Test Statistic	Significance *
NH-SSH	−1.262	5.540	−0.228	0.820	1.643	5.540	0.297	0.767
NH-HWS	4.310	5.540	0.778	0.437	9.357	5.540	1.689	0.091
NH-ISH	−6.119	5.540	−1.104	0.269	8.500	5.540	1.534	0.125
NH-ISHI	−13.119	5.540	−2.368	0.018	19.357	5.540	3.494	˂0.001 *
SSH-HWS	−5.571	5.323	−1.047	0.295	−7.714	5.323	−1.449	0.147
SSH-ISH	4.857	5.323	0.912	0.362	−6.857	5.323	−1.288	0.198
SSH-ISHI	11.857	5.323	2.228	0.026 *	−17.714	5.323	−3.328	0.001 *
HWS-ISH	10.429	5.323	1.959	0.050 *	0.857	5.323	0.161	0.872
HWS-ISHI	17.429	5.323	3.274	0.001 *	−10.000	5.323	−1.879	0.060
ISH-ISHI	−7.000	5.323	−1.315	0.188	10.857	5.323	2.040	0.041 *

NH: normal hip joints; ISH: instability surgery hip joints; SSH: sham surgery hip joints; ISIH: instability surgery hip joints with pelvic limb immobilization; HWS: hip joints without surgery; * Groups revealed statistical significance differences.

## Data Availability

The raw data supporting the conclusions of this article will be made available by the authors upon request.

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
