# Peer review of "Morphometric Assessment of the Hip Joint in a Functional Dysplastic Rabbit Model"

_vetsci, 2024, doi:10.3390/vetsci11080387_

Round 1

Reviewer 1 Report

Comments and Suggestions for Authors

This manuscript provides a morphometric assessment of the coxofemoral joint in a functional hip dysplasia rabbit model. As the title suggests, the manuscript is clearly focused on the morphometry of the hip joint and present a large dataset of multiple parameters that were determined by means of radiography and CT-scanning.

Simple summary

-       Line 13 and 14: Please mention what animal species you are targeting. For what species could your model be valuable? Now, it is first mentioned in line 17. Add such information to the introduction too, in which no target species is mentioned.

-       - Line 19: Remove “In conclusion”

Abstract

-       Line 28-29: “particularly in the group of hip instability surgery with hindlimb immobilization”. So, you suggest that multiple types of surgery were performed. However, line 24 mentions that only the round ligament was transected. Please add this info to the abstract.

-       Line 29-30: “The results highlight the critical influence of hip joint instability on the development of hip dysplasia”. Could it also be the other way around? 

Materials and methods:

-       Line 81: Mention already here what the purpose of the stifle fixation in extension is.

-       Line 126-127: Something went wrong here with the positioning of the text.

Results

-       General remark: the symbol for degrees is °.

-       Line 296: “at the time of sacrifice” You have not mentioned and described the sacrifice of the animals in the Materials and methods. 

-       Line 322-324: Please rephrase this sentence, as it is grammatically wrong.

-       Line 329-330: Table 1 is mispositioned within the text. The heading of the table must be positioned above and not below the table.

-       Both tables: Provide the units for the values mentioned in the tables.

Figure 1: “a” is floating outside the figure and “b” is nowhere to be seen.

Discussion:

-       General remark: As this paper will certainly be read by investigators interested in the broader concept of hip dysplasia, it would be valuable to include more data on the comparison between hip dysplasia in the rabbit and other species prone to this disease. Also elaborate on how to use the rabbit model to study hip dysplasia in other species. Thus, try to answer the question “What can we do with your paper?”.

Conclusions

-       Line 413-414: “The data of our study may be used in future studies performed in rabbits with naturally occurring or hip dysplasia-induced disease.” Are you suggesting that your model is valuable to study hip dysplasia in the rabbit? You are not mentioning the rabbit as a model to study hip dysplasia in humans and dogs. It is not clear for what species the rabbit is a model. Also see my general remark for the Discussion.

Comments on the Quality of English Language

Some sentences seem to be quite literally translated from the native language. They are understandable, but a native English speaker would definitively rephrase these.

Author Response

Dear reviewer,

Due to technical issues consider only the revision in the file uploaded. 

Reviewer 2 Report

Comments and Suggestions for Authors

The reviewed paper is good contribution to the anatomy and physiology of locomotor system and contain new findings valuable also for the physiotherapy and animal orthopedy. The work was well organized and written, but some less important editorial remarks were indicated in text. The main problem is nomenclature, which is not corresponding with Nomina Anatomica Veterinaria. I hope, Authors are able to correct these issues indicated in pdf file. The study indicates and discusses the problems arising from findings in this field of knowledge. The results allows for forming the new approaches, which can be taken under consideration in further studies. Introduction gives good view on general subject of investigation. Material and methods – methods are clearly described; Discussion is well written and drive us to the conclusions, but maybe wider description of hip joint and femur morphology and differences between rabbit – dog and human would be beneficial. Moreover I suggest to add some information about the sensoric innervation of joint capsule, which plays an important role in symptomatology. I suggest to revise carefully the whole text, my remarks are indicated in pdf file. I suggest to accept this work after minor revision.

Author Response

(The authors gave the same response as above.)

Reviewer 3 Report

Comments and Suggestions for Authors

The article is original and very relevant for the field. The authors studied investigates the morphometric changes in the coxofemoral joint in a surgical-induced rabbit model of hip dysplasia through the sectioning of the round ligament.

The results showed revealed that the surgery caused significant changes in the hip, making it similar to the hip dysplasia seen in dogs and humans. These results highlight the critical influence of hip joint instability on the development of hip dysplasia. These findings provide a foundation for future research on naturally occurring or experimentally induced hip dysplasia in rabbits and underscore the model's potential for studying the biomechanical and developmental aspects of hip joint disorders.

The methology of the study is modern and reproducible, authors providing detailed and very accurate imagistic.

The conclusions are consistent with the evidence and arguments presented.

The references are appropriate, including some relevant authors experience in the field.

I recommend some minor corrections.

-line 34- Keywords, the first should be hip dysplasia

-line 126- delete the femur [10]

-line 329- Title of Table 1 should be moved to line 314

Line 331- legend of Table 1 should be moved under the Table

Reference 4 is incomplete- give detailes of the article published

Ref 10, 19, 22 are incomplete- give the issue and pages or article nr

Author Response

(The authors gave the same response as above.)

Reviewer 4 Report

Comments and Suggestions for Authors

This study examined the anatomical change of the coxofemoral joint after dissection of the round ligament. The treatment of animals and the procedure of animal experiments in this study are appropriate, and results demonstrated in this study are attractive and useful for the anatomist and the surgeon. However, there is one concern and some mistakes in this manuscript and appropriate revision should be required.

My major concern is the point why the statistical significance could not be observed between ISH-ISHI for femoral anteversion angle. The result of this study demonstrated that surgery with hindlimb immobilization is important to elicit anatomical abnormality in yong rabbit hip joint. I want to the authors to describe the reason for non-statistical significance between ISH-ISHI, in the revised manuscript.

Minor comments:

1. In line 127, “the femura [10]” should be deleted.

2. “a” and “b”, those are important landmarks, could not be found in figure 1.

3. The mark of significance “*” should be put near the corresponding values in table 2.

4. Please check the journal name in the references. Some are shown as abbreviations name, but others are shown as full name.

Author Response

(The authors gave the same response as above.)

Round 2

Reviewer 1 Report

Comments and Suggestions for Authors

Dear authors,

Thank you for revising your manuscript according to the suggestions made by the reviewers. I have no further comments.